# Biomimicry and 3D-Printing of Mussel Adhesive Proteins for Regeneration of the Periodontium—A Review

**DOI:** 10.3390/biomimetics8010078

**Published:** 2023-02-12

**Authors:** Jan C. Kwan, Jay Dondani, Janaki Iyer, Hasan A. Muaddi, Thomas T. Nguyen, Simon D. Tran

**Affiliations:** 1McGill Craniofacial Tissue Engineering and Stem Cells Laboratory, Faculty of Dental Medicine and Oral Health Sciences, McGill University, Montreal, QC H3A 0C7, Canada; 2Department of Oral and Maxillofacial Surgery, King Khalid University, Abha 62529, Saudi Arabia; 3Division of Periodontics, Faculty of Dental Medicine and Oral Health Sciences, McGill University, Montreal, QC H3A 0C7, Canada

**Keywords:** mussel adhesive protein, biomimetics, periodontium, 3D printing, polydopamine, dentistry, peri-implantitis, periodontitis, tissue engineering, biomaterials, implant dentistry

## Abstract

Innovation in the healthcare profession to solve complex human problems has always been emulated and based on solutions proven by nature. The conception of different biomimetic materials has allowed for extensive research that spans several fields, including biomechanics, material sciences, and microbiology. Due to the atypical characteristics of these biomaterials, dentistry can benefit from these applications in tissue engineering, regeneration, and replacement. This review highlights an overview of the application of different biomimetic biomaterials in dentistry and discusses the key biomaterials (hydroxyapatite, collagen, polymers) and biomimetic approaches (3D scaffolds, guided bone and tissue regeneration, bioadhesive gels) that have been researched to treat periodontal and peri-implant diseases in both natural dentition and dental implants. Following this, we focus on the recent novel application of mussel adhesive proteins (MAPs) and their appealing adhesive properties, in addition to their key chemical and structural properties that relate to the engineering, regeneration, and replacement of important anatomical structures in the periodontium, such as the periodontal ligament (PDL). We also outline the potential challenges in employing MAPs as a biomimetic biomaterial in dentistry based on the current evidence in the literature. This provides insight into the possible increased functional longevity of natural dentition that can be translated to implant dentistry in the near future. These strategies, paired with 3D printing and its clinical application in natural dentition and implant dentistry, develop the potential of a biomimetic approach to overcoming clinical problems in dentistry.

## 1. Introduction

As technological innovation continues to push the limits of what is possible in the healthcare field, nature has played an important role in addressing human problems and challenges for thousands of years. Humans have always drawn inspiration and guidance by studying and emulating designs, processes, and strategies based on solutions proven by nature. This practice can be referred to as “Biomimicry”, which involves understanding the way natural systems work and using that understanding to create new technologies and materials that mimic their behavior, structure, or function. The practice of biomimicry can be dated as far back as the start of human civilization, with one of the earliest known records of biomimicry being silk fabric from the silkworm. Biomimicry has always played an influential and important role in human development as it allows for more efficient and effective technologies that can also lead to materials and systems becoming more biocompatible and sustainable. The application of biomimicry also spans a wide range of fields, including dentistry, architecture, renewable energy, biomechanics, materials sciences, and microbiology [1].

The use of biomimicry in dentistry has been extensive, with the development of dental materials that have been researched to be resistant to degradation and biocompatible, such as hydroxyapatite (HA), casein-phosphate, and bioactive glasses [2,3,4]. Other developments include an implant design involving different surface properties to improve its stability and tolerance in vivo, such as plasma spray, bioactive components, and biomimetic surface patterning [5,6,7]. These advances in the field of dentistry have fueled more recent innovations in the combination of tissue engineering, regeneration, and replacement in combination with 3D printing. Biomimicry, within the field of tissue engineering, is often used to design and develop 3D structures or scaffolds that can aid in the growth and development of living cells. The design of these structures mimics the structure and function of natural dental tissues and can be used to repair damaged or diseased tissues in the oral cavity. Additionally, 3D printing or additive manufacturing adds another layer to this concept, with the ability to control all aspects of the design phase to ensure the precise fabrication of the complex scaffold or 3D structure at a micron level. A key requirement for a functional scaffold is a 3D environment or matrix that mimics an extracellular matrix (ECM) for cells to be seeded. This enables a medium for cellular functionality, growth, and assembly [8].

A major focus has now shifted to using the advantages of atypical 3D-printed biomimetic biomaterials to mimic the function of the periodontal ligament (PDL) in the periodontium through means of tissue engineering, regeneration, and replacement. The PDL consists of a group of fibrous tissues that attach teeth to the surrounding bone in the jaw. These tissues play a critical role in maintaining the stability of natural dentition and allowing them to function properly. However, with the current prevalence of infection in the oral cavity, such as periodontitis in natural dentition and peri-implantitis in dental implants, many recent studies have elucidated potential preventative solutions to either reverse or reduce the effects of both periodontitis and peri-implantitis using different biomimetic biomaterials for the purpose of PDL engineering, regeneration, or replacement [9,10]. In this review, we highlight an overview of the application of different biomimetic biomaterials in dentistry and discuss the key approaches that have been researched as biomimetic solutions for periodontitis and peri-implantitis. Following this, we focus on the recent novel application of mussel adhesive proteins (MAPs) and their key chemical properties that relate to the engineering, regeneration, and replacement of the PDL. Finally, these strategies, paired with 3D printing and their clinical applications in natural dentition and implant dentistry, are proposed, and we outline the potential of a biomimetic approach to overcoming clinical problems in dentistry.

## 2. Biomimetic Biomaterials in Dentistry

The use of biomaterials in the fields of medicine and dentistry has encompassed a range of widely researched and applied concepts since the 1950s [11]. The advent of tissue engineering demonstrates the relentless quest for materials that mimic the human microenvironment in order to enhance tissue repair and regeneration in structure and function [12].

Mechanical considerations, such as the load bearing, fracture resistance, tensile strength, hardness, modulus of elasticity, and wear resistance to abrasion and corrosion, all play a vital role in the ideal function of biomaterials [13,14] Similarly, the wear resistance of biomaterials is of great importance to avoid the release of wear debris into the surrounding tissues, which may trigger a foreign body reaction. Multiple or cyclic stresses incorporated during the processing, finishing, and prolonged use of biomaterials in vivo can cascade into chronic or long-term fatigue. The technical, safety, and economical specifications are dictated by the intrinsic and extrinsic properties of the material [14]. The intrinsic properties include the material bulk, surface physicochemical properties (atomic number, chemical structure, and density), mechanical properties (elastic modulus, yield stress, and fracture stress), and other properties that are independent of the quantity of material matter used [14]; extrinsic properties highlight the dimensional factors (size and shape), material, and production costs [14].

The natural environment of the human body is harsh and can expose biomaterials to varying levels of pH and humidity from bodily fluids, which may result in corrosive and toxic responses owing to inadequate corrosion resistance and degradation of the biomaterial [14]. In addition to adverse host reactions such as inflammatory or immunological rejection, a biomaterial implanted into the body must conform to the inside harsh physiological environment for longevity. Biocompatibility signifies the crucial aspect of material science that involves the ability of the material to be used in approximation with living tissue, without causing any toxicity, adverse inflammatory or allergic reactions, carcinogenic or mutagenic action [13]. Prolonged human contact with such substances advocates a stringent selection of properties, such as biocompatibility, mechanical compatibility, abrasion and corrosion resistance, toxicity, long fatigue life, and cost-effectiveness [13,15]. The use of biomaterials inspired by nature (biomimetic materials) has recently been advocated to solve many of the aforementioned complications.

Biomimetic biomaterials, as the name suggests, are biomaterials which mimic the tissues of the body anatomically and physiologically to generate an effect similar to that of the native tissues. Numerous materials that mimic the constituents of the periodontium, particularly bone, are being used for the regeneration of the periodontium. Due to its favorable biological activity and biocompatibility, HA is commonly used as a scaffold in artificial bone [16]. The consideration of HA is accentuated by the fact that, in addition to being osteoconductive and osteoinductive, the material possesses the same structure as human bone [17]. Although there are differences in the bioactive and mechanical properties of bone and HA, researchers have been able to formulate HA-polymer composites that increase its toughness and degradation rate. Scaffolds made from these types of materials have been reported to be ideal for bone tissue engineering [18,19]. Additionally, surface modifications with apatitic compounds, such as HA, can be used to improve the primary stability of dental implants in the bone [18,19]. Other examples include bioactive glass, which has been used to regenerate hard and soft tissues due to its ability to bond to both types of tissues and precipitate HA in aqueous solutions [20]. Bone morphogenetic protein-2 (BMP-2) plays an important role in the development of cartilage and bone, which has been extensively researched in the context of the hard tissue regeneration of the periodontium [21]. The development of new biomimetic biomaterials in dentistry is constant and has showcased the current advancement of the field in mimicking natural oral tissues (Table 1).

## 3. Biomimetic Biomaterials in Periodontium Regeneration

The periodontium is a group of tissues (gingiva, PDL, cementum, and alveolar bone) that support and keep the teeth in place. Each one of these tissues has unique histological and biological characteristics [22]. Periodontitis is a chronic inflammatory condition that destroys the periodontium, resulting in tooth-loss and severe functional and aesthetic issues for patients. Almost 50% of the global adult population suffers from this condition. Regardless of the periodontitis’ etiology, the initial onset begins with inflammation, which is a complicated mechanism that interferes with the periodontium’s turnover and repair processes [23].

Investigations into the use of conventional regenerative methods, such as guided tissue regeneration (GTR), have been limited and unpredictable in clinical trials. This surgical approach works by reflecting the mucoperiosteal flap, removing the causative causes, and blocking the defective area with biocompatible biomedical materials placed as a barrier under the gingiva to prevent epithelial tissue migration [23,24]. The current standard for GTR is the biomaterial polytetrafluoroethylene (PTFE), which is a non-absorbable biomaterial with excellent physical and antibacterial attributes. Although it has been reported to show great results in periodontal regeneration, the need for a second surgery may disturb the healing process and increase the risk of infection [25,26]. Even with the improvement of the existing biomaterials, GTR membranes are insufficient to achieve the desired therapeutic results. GTR membranes may be more effective in tissue regeneration when used in conjunction with drugs carried by the membranes and other methods, such as bone grafts [24]. Thus, the regeneration of the structures and functions of the periodontium in patients with periodontitis necessitates the development of alternative regeneration techniques.

Tissue engineering is another periodontal regeneration strategy. In order induce tissue regeneration, stem/progenitor cells, scaffolds, and bioactive agents must be used to create biomimetic systems. This technique can be classified, according to its biomaterial usage, as either scaffold-free or scaffold-based [24]. Scaffold-based tissue engineering was pioneered in order to emulate the natural 3D structure of the ECM and duplicate the periodontal microenvironment in the tissues around the teeth. Furthermore, the biocompatibility and cell affinity of the scaffolding materials are crucial for avoiding immunological reactions and promoting tissue regeneration [27,28]. The alveolar bone and cementum are hard tissues; thus, the scaffold for hard tissue regeneration should promote mineralized tissue production. In contrast, PDL is a fibrous tissue; therefore, the scaffold for PDL regeneration should promote soft tissue formation while preventing mineralization. Inorganic biomaterials, such as HA and calcium phosphate, are employed as scaffolding components for mineralized tissues, whereas polymeric biomaterials are commonly used for PDL regeneration. The periodontium is a complicated structure consisting of a cementum-PDL-alveolar bone complex that are all interconnected, making the regeneration of only one layer of tissue challenging. To mimic the periodontal structures, multi-layered scaffolds with different compositions, structures, and architectures in each phase are necessary [24,29]. Recent advancements in 3D printing technology have allowed for a new perspective on the design and application of scaffolds, which can precisely offer more biophysical cues, such as controlled porosity within the scaffolds that replicate the natural 3D ECM of a defected tissue to promote tissue regeneration. In addition to using cone beam computer tomography (CBCT) scanning, a 3D printed scaffold may be tailored according to the specific needs of patients. Despite its usefulness, the 3D printing technique is limited and cannot be used to create nanoscale scaffolds that match the ECM’s architecture (Figure 1) [24,25,26].

Another highly important aspect of periodontium regeneration is controlling the release patterns of drugs and growth factors from carriers to obtain sustained long-term effects while avoiding negative biological effects. Therefore, periodontal tissue regeneration requires the administration of a combination of bioactive chemicals to keep infection and inflammation under control, while boosting cell proliferation and differentiation [24,30]. Recently, a novel immunomodulatory method has been developed to promote the regeneration of the alveolar bone. Macrophages, which are crucial in both the initiation and resolution of inflammation, were the focus of this strategy. Macrophages are typically classified as either a pro-inflammatory M1 or anti-inflammatory M2 phenotype depending on their activation state. Based on their surrounding microenvironment, macrophage phenotypes can switch rapidly. Interleukin 4 (IL-4) is a potent cytokine that can switch the M1 to the M2 phenotype. Therefore, binding IL-4 to a nanofiber’s heparin-modified gelatin microsphere showed excellent regulation of inflammation and bone regeneration in vivo [30,31].

A cell-sheet approach, which is a novel scaffold-free tissue engineering method that transplants cultured stem/progenitor cells without carriers, has only been able to regenerate simple structures [27,29]. Cementum is a thin mineralized tissue formed around that tooth for PDL attachment. Studies showed that scaffolds designed for bone regeneration do not work well in cementum regeneration [25]. Culturing periodontal ligament stem cells (PDLSCs) and bone marrow mesenchymal stem cells (BMSCs) in dishes have shown the ability to differentiate into osteoblasts, fibroblasts, and cementoblasts. Moreover, a thin layer of mineralized cementum-like tissue was generated on the root surface as the cell sheets expressed the CEMP1 signal. Therefore, regenerating the cementum-PDL-bone complex requires combining the cell sheets with the PDL and bone scaffolds [32]. Cell-homing, gene therapy, and multilayer materials for periodontal regeneration still require further in vivo and clinical research. The success of PDL, cementum, and alveolar bone regeneration, as well as other fields of tissue engineering and regenerative medicine, may be significantly impacted by these current findings on stem cells and innovative scaffolds [23].

## 4. Mussel Adhesive Proteins in Dentistry

MAPs have been an exciting avenue as a biomimetic biomaterial in the field of dentistry. MAPs are adhesive secretions from the mussel foot of the *Mytilus genus*, which they can use to attach themselves to many different inorganic and organic surfaces within their marine environments. MAPs consist of as many as 20 different proteins that entail some collagenous and adhesive plaque proteins that all work together in the mussels’ adhesive process, which consists of a series of byssal threads that anchor the organism to its substrate. One of the key chemical properties of these MAPs is their high catechol group content, consisting of a six-membered aromatic ring with two hydroxyl groups attached (Figure 2). These catechol groups are highly reactive and can form strong chemical bonds with a variety of surfaces; however, the specific catechol group that is known to be responsible for the unique adhesive properties of MAPs is 3,4-dihydroxy-L-phenylalanine (DOPA), which is a post-translationally modified amino acid of tyrosine. There is still little understanding of the exact physiochemical details of DOPA-substrate interactions and the role of DOPA in MAPs.

There have been many studies that have elucidated the importance of DOPA, and in particular, their potential application in cross-linking reactions of catechol groups via oxidation to aid in the rapid solidification of concentrated solutions of DOPA for attachment onto substrates [33]. There is also evidence that the adhesive strength of MAPs can be linked proportionally to the DOPA concentration [34]. Although there is vast research demonstrating the potential biomedical applications of MAPs, there are inherent drawbacks to their use. The direct extraction and isolation of MAPs from the mussel secretory gland is labor-intensive and costly, and the number of mussels that are required to produce only 1 g of MAP is estimated to be in the thousands [35,36]. Moreover, the surrounding pH environment also plays an influential role in the adhesive nature of MAPs as, in a neutral or basic pH, DOPA can readily oxidize [33]. Despite these challenges, successful attempts have been made to produce functional and economical mussel-inspired protein recombinants, which mimic the adhesive characteristics of the natural MAPs found in mussels, that can be commercially produced [35,36,37]. One example of a successful attempt is a study by Bolghari and colleagues that investigated a novel combinational bio-adhesive chimeric protein that included the fusion of the gas vesicle protein A (GvpA) of *Dolichospermum fosaquae*, prokaryotic curli protein CsgA of *Escherichia coli* (*E. coli*), and DOPA-containing mussel foot proteins 3 and 5 (Mfp3, Mfp5), all expressed in methylotrophic yeast, *P. pastoris* [37]. The authors concluded that *P. pastoris* a suitable expression system of adhesive proteins, allowing for a higher production rate and improved post-translational modifications for the function of the protein, in addition to the greater cohesion of proteins in acidic pH 3–5. Successful attempts have also been made to produce mussel-inspired synthetic polymers, such as the more notable Polydopamine (PDA), through a simple dip-coating method in an aqueous solution of dopamine [38]. The implications of these studies allow for a plethora of potential biomedical applications, particularly in tissue engineering in dentistry, due to the challenges and environment of the oral cavity.

As the technology of bioadhesives improves, their applications in the dental field have become an important step in replacing the deteriorating key anatomical structures of the periodontium, that include periodontal tissues to teeth or regenerating surrounding bone loss near dental implants, in cases of periodontitis and peri-implantitis, respectively. The current research has been primarily focused on the applications of functionalized PDA with respect to guided bone regeneration (GBR) and GTR paired using notable Food and Drug Administration (FDA) cleared biocompatible and biodegradable 3D polymer scaffolds, such as polylactic acid (PLA) or poly(ε-caprolactone) (PCL) [39,40,41]. The reasoning behind this pairing is that the use of polymers such as PLA and PCL as 3D scaffolds can allow for surface functionalization and is suitable for cell adhesion; however, traditional means of incorporating genetic material or proteins within these polymers during the process of 3D printing at high temperatures have created challenges, such as degradation or denaturation. As a result, the use of PDA coatings has provided an alternative means to adhere genetic materials onto 3D-printed scaffolds and improve their bioactive potentials.

In one study, investigators examined a PCL-based nanofibrous membrane scaffold with and without an incorporated biomimetic PDA nanoscale coating. The reported results demonstrated that using a patterned PDA coating allowed an adherent substrate for cells, but also induced the osteogenic differentiation of cultured PDLSCs [39]. In another study, the concept of PDA coating was further explored in the attachment of pig cornea tissues onto three different common implantable materials: aluminum, polymethyl methacrylate (PMMA), and titanium (Ti) [42]. The investigators concluded that using a gelatin hydrogel crosslinked with microbial transglutaminase and an implantable material’s surface coated in PDA increased adhesion between the implant surface and pig cornea tissues as compared to non-PDA coated surfaces. Li and colleagues investigated periodontal bone regeneration under diabetic inflammatory periodontal conditions [43]. The investigators were able to take advantage of PDA’s antioxidant properties and reactive oxygen species scavenging ability to develop PDA-reduced graphene oxide and PDA-modified HA nanoparticles incorporated into a conductive alginate/gelatin scaffold. This PDA functionalization strategy onto graphene oxide and HA nanoparticles aided in their dispersibility in the alginate/gelatin network and improved the conductivity of the graphene oxide, allowing for a conductive pathway to signal and activate the Ca^2+^ channels. The results show that the scaffold was able to facilitate cell adhesion and, with the help of the catechol groups of PDA, confer an anti-inflammatory effect and accelerate periodontal bone healing under diabetic conditions.

As the technology and clinical application of 3D-printed scaffolds improve within dentistry, an important consideration that must be acknowledged is 3D printing or additive manufacturing. Although 3D printing has become very useful in creating complex scaffold shapes, they are not without their limitations (Table 2). Some of these limitations of 3D-printed scaffolds include potential internal scaffold defects, dimensional inaccuracies, poor mechanical properties, printing speed, post-print processing, and cost [44,45]. The importance of maintaining the scaffold’s internal structure is vital for its biological capabilities; therefore, certain parameters are required to allow for the most dimensionally accurate 3D-printed scaffold model, relative to any given defect. Currently, many recent studies have shed light on these necessary parameters in the context of 3D printed scaffolds. Holzomond and Li developed a measuring technique to monitor, in real-time, the surface geometry of a printed part using 3D digital image correlation (3D-DIC), enabling the detection and location of defects in the 3D-printing process [46]. Hawker and colleagues combined the use of computer-aided design (CAD) and finite-element modelling (FEM) prior to 3D printing and were able to test the printability limitations, thus improving the predictability of 3D printed scaffolds in relation to size, geometry, and pore size [47]. Zhao and colleagues took an alternative parametric approach to assess how different geometric variations in the scaffold architecture, pore size, and porosity responded to mechanical stimulation using computational fluid dynamics (CFD) and fluid-structure interaction (FSI) models [48]. The ability to model, analyze, and optimize 3D scaffolds prior to printing through parametric analysis introduces a necessary step in the workflow of 3D-printed scaffolds. This ensures the monitoring and elimination of any potential defects that may compromise the functionality and efficacy of 3D-printed scaffolds.

PDA coatings have also been extensively demonstrated to enhance the expression of bone-related genes, osseointegration, cell adhesion and proliferation, and aid in periodontium regeneration. Another unique aspect of PDA coatings that extends their application beyond the periodontium is their ability to possess antimicrobial activity in conjunction with their existing properties. Peri-implantitis has become a topic of discussion due to its ambiguous etiology, with multiple theories that propose a biological, genetic, and biomechanical mechanism that results in the onset of peri-implantitis. Currently, the exact mechanism behind peri-implantitis is still not entirely understood. One study approached the problem of peri-implantitis from a biological approach, detailing biofilm formation and bacterial colonization as contributing factors [49]. Investigators developed a biomimetic multilayer consisting of Ti–6Al–4V—Silver (Ag)—PDA and found that placing PDA as the top layer protected the Ag from corrosion and allowed for long-term sustained release, along with antibacterial activity against *E. coli* and *Streptococcus aureus* (*S. aureus*). Another study investigated the immobilization of dopamine and cefotaxime sodium (CS) onto the surface of Ti via copolymerization, which was able to prevent the adhesion and proliferation of *E. coli* and *Streptococcus mutans* (*S. mutans*) [50]. Mendhi and colleagues synthesized a PDA-based copper coating onto Ti that can regulate endogenous nitric oxide generation, leading to significant reductions in the biofilm metabolic activity and bacterial attachment paired with copper’s antimicrobial properties [51]. The combination of PDA and targeted drug delivery to regulate inflammation while regenerating tissue or bone highlights the importance of these studies and the various tools that can potentially be used to target key structures in the periodontium.

A common theme among many in vitro experiments in the field of bioadhesives, particularly the use of MAPs and PDA, is that they are primarily used as surface coatings for substrates or 3D scaffolds to help immobilize various proteins or genetic materials of interest (Figure 3). Very few studies have investigated using PDA as a concentrated bioink to 3D-print a stand-alone scaffold or microstructure that can potentially relate to the specific needs within the field of dentistry. Im and colleagues successfully developed a bioink consisting of alginate, tempo-oxidized cellulose nanofibrils, and PDA nanoparticles, which was capable of being 3D-printed and inducing osteogenic differentiation [52]. Luo and colleagues combined the use of concentrated alginate/polydopamine inks to create a 3D-printed biphasic scaffold and a cell-laden hydrogel that can change shape when triggered by near-infrared light [53]. Both of these studies outline the potential of using a PDA bioink for 3D printing or additive manufacturing that can meet the specific needs of GBR and GTR in the dental field. Additionally, the implications of these studies also introduce another, similar, technology to 3D printing, known as bioprinting. Bioprinting utilizes the 3D printing technique but instead prints with cells and biomaterials that can produce organ-like structures that can imitate the characteristics of natural tissues. Currently, marine-animal-derived biopolymers have gained significant interest as bio-inks for bioprinting due to their favorable bioactive potential for tissue engineering applications, while being well-tolerated in vivo [54,55]. Commonly researched marine-derived biopolymers include alginate, chitosan, collagen, and HA, which have been investigated for targeted drug delivery, cell scaffolds, and bone or tissue regeneration (Figure 4) [54,55]. However, the current limitations of many of the marine-derived biopolymers are their controlled degradation in vivo and their weak mechanical properties post-printing [56]. However, one avenue that has been proposed to overcome these challenges is the use of nacre, found in the inner layer of mollusk shells, which have been extensively researched as a model for dental and bone applications [57,58]. Studies have begun characterizing the favorable mechanical properties of nacre by developing nacre-like materials that could be incorporated into the bioprinting process [59,60,61]. One of the advantages of nacre is its hard aragonite nanocrystals and soft biopolymer matrix, which possesses the ability to strengthen itself during deformation [62,63]. This leads to the potential application of nacre-like materials, combined with other marine-derived biopolymers, to improve the current mechanical limitations of 3D scaffolds via bioprinting for periodontium regeneration within dentistry. Further research in vitro and in vivo will be required to formulate the best composition of bio-inks that could be implemented in an economical way while maintaining high bioactive potential, particularly for cell scaffolds.

The key chemical properties of MAPs have been utilized effectively by many researchers to improve and develop novel bioadhesive technologies in the realm of tissue engineering. PDA has become fundamental in immobilizing the key proteins, genetic material, antibiotics, and inorganic and organic antibacterial agents. However, a complete understanding of MAPs and PDA-mediated techniques are still required, as the exact mechanisms of their physiochemical properties remain elusive. Further research should be conducted on the interactions between PDA-coated substrates and cells, including tissue stem cells and bone-resorbing mediators. The application of MAPs in tissue engineering has become increasingly widespread and it will be important that the prior clinical application of these novel bioadhesives undergo rigorous human clinical trials to ensure their safety and efficacy in the short- and long-term.

## 5. Clinical Application of MAPs in the Periodontium

Due to MAPs’ distinct ability to retain their adhesive properties in a moist environment, they can also be employed in various clinical procedures that require the proximity of adherends. The anti-microbial properties of MAPs and the increased differentiation of osteoblasts by MAPs make them more suited as biomaterials for restoring the periodontium.

MAPs, as a surface coating for implants, have been theorized and evaluated for the facilitation of osseointegration. A statistically significant difference in the differentiation of bone marrow stem cells (assessed by alkaline phosphatase activity and alizarin red staining) has been observed with the MAP coating of implants [64]. Moreover, biomimetic mineralization with simulated body fluid demonstrated significantly more mineral crystals with a MAP surface coating. Blending MAPs and gelatin to be loaded into a nanotube Ti dental implant has also been reported to enhance osseointegration [65]. Mussel-inspired PDA coatings have also been reported to efficiently encourage the immobilization of BMP-2 on Ti surfaces, with the modified Ti substrate demonstrating an enhanced osteogenic differentiation of PDLSCs through the integrin-mediated cell-matrix adhesion mechanisms [66]. When compared to a control group, Yin and colleagues concluded that MAP coatings on Ti with nanonetwork structures increased the bone-implant contact [67]. These findings substantiate the potential of MAPs as an implant surface coating. MAP-coated implants may find application in cases where the prognosis is poor due to anatomical causes, for example, for buccal wall resorption in conjunction with GBR. MAPs may also serve as an implant coating for cases where systemic skeletal and metabolic conditions decrease the quality and quantity of a host’s bone for implantation. The efficacy and effectiveness of MAP implant coatings have been validated, but the problem of efficiency has not been assessed. Further evidence will be required to establish MAPs as an efficient implant coating for cases where conventional implant coatings are ineffective.

Due to their tenacious nature, MAPs have also been considered in GBR as an adjunct to conventional bone grafts. The effect of mussel-inspired PDA coatings on calcium silicate cement was studied on mesenchymal stem cells. The coating enhanced cell adhesion and promoted ECM secretion, including collagen I and fibronectin. In addition, there was a statistically significant increase in cell-adhered proteins, such as integrin f1 and pFAK, which resulted in accentuated cell proliferation [68]. Additionally, PDA coatings have been demonstrated to stimulate osteogenesis and the differentiation of cells [68,69]. Thus, MAPs can potentially be utilized with bone grafts to increase their osteoinductivity. The role of MAPs as a pre-treatment of bone grafts may be significant in cases where bone regeneration is required outside the osseous contour due to anatomical defects. MAP hydrogels may be used in conjunction with platelet-rich plasma (PRP) or platelet-rich fibrin (PRF) in GBR. However, at present, there is a paucity of evidence of the effectiveness of this combination and will require in-vitro studies with simulated body fluid (SBF) for substantiation. Moreover, further in-vivo studies will be required to establish MAPs as a reliable pre-treatment for bone grafts.

MAPs also improve the handling characteristics of GBR membranes as the need for secondary fixation devices (bone screws or bone tacks) or techniques (periosteal suturing) is eliminated. However, MAPs have not been shown to increase the rate of bone regeneration. The use of tissue adhesives may cause resorbable collagen membranes to degrade prematurely [39]. An improvement in the compatibility of the GBR membranes with MAPs is required to increase the resistance of the membranes to resorption. Meanwhile, the performance of MAP-coated synthetic membranes, such as PTFE, should be evaluated, as these types of synthetic membranes are more resistant to degradation.

MAPs can also be used as adhesives for the primary closure of wounds. Mussel foot proteins (MFPS)-inspired double cross-linked hydrogel adhesives have been assessed for use as a potential adhesive for biomedical applications. Subcutaneous implantation demonstrated the biodegradability and biocompatibility of the double cross-linked hydrogel [70]. The application of MAPs as wound closure adhesives is yet to be investigated and is quite novel, but it has the potential to replace conventional modalities, such as suturing and the application of collagen tapes.

MAPs may be used as an adhesive medium for local drug delivery (LDD) in sites of incipient periodontal lesions. Similarly, MAPs may be employed for the retention of periodontal dressings following surgical procedures, such as flap or mucogingival surgery. Extensive in vitro and clinical studies will be required for the validation of these applications. At present, the utilization of MAPs to regenerate and replace key periodontium structures has been extensively studied, with the same techniques reported in these studies, the application of MAPs can also extend into the field of implant surface treatments and bone graft coatings with immense potential, but will require human studies for commercialization.

## 6. Conclusions

The current advancements in bioadhesive technology have helped provide an extensive understanding of the mechanism by which MAPs and PDA adhere to many different substrates. This has resulted in 3D scaffolds with increased bioactive potential and PDA-functionalized coatings that can initiate bone and tissue cellular differentiation. Preliminary research has exploited these biomimicking properties to enhance tissue and bone repair and regeneration in the periodontium, thus serving as a promising therapeutic measure. Although it mandates substantial experimentation and clinical trials, further applications of MAPs in the fields of implant dentistry remain dynamic.

## Figures and Tables

**Figure 1 biomimetics-08-00078-f001:**
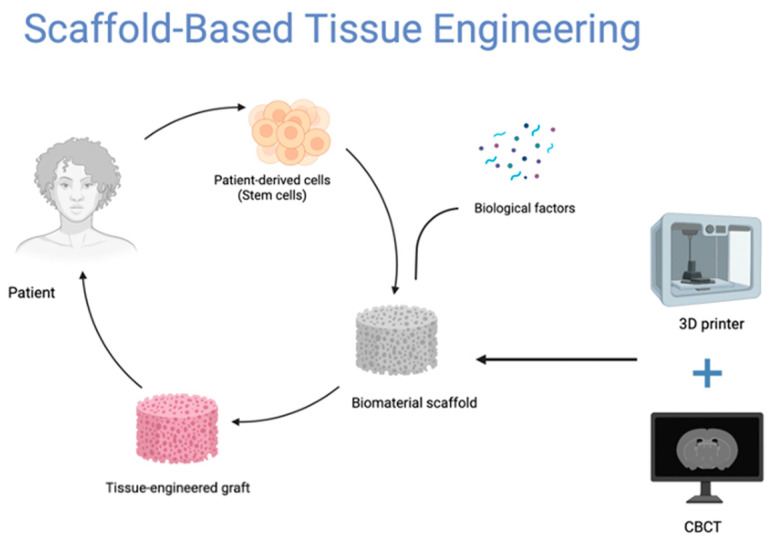
Schematic illustration of the incorporation of CBCT scanning and 3D printing in the workflow of scaffold-based tissue engineering.

**Figure 2 biomimetics-08-00078-f002:**
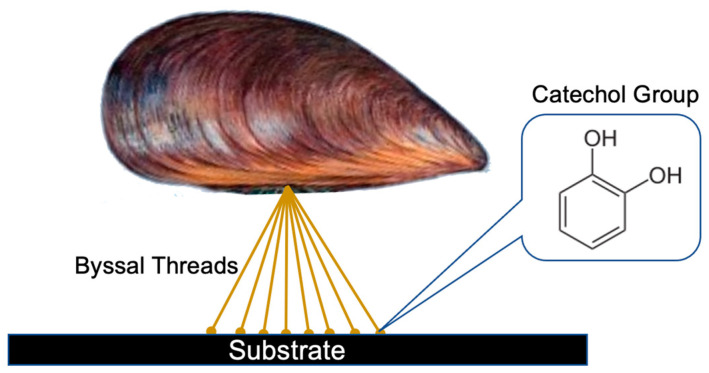
Schematic representation of a mussel using its byssal threads to attach to the surface of a substrate. The key chemical composition of its byssal threads is catechol groups which make up MAPs and their unique strong adhesive characteristics.

**Figure 3 biomimetics-08-00078-f003:**
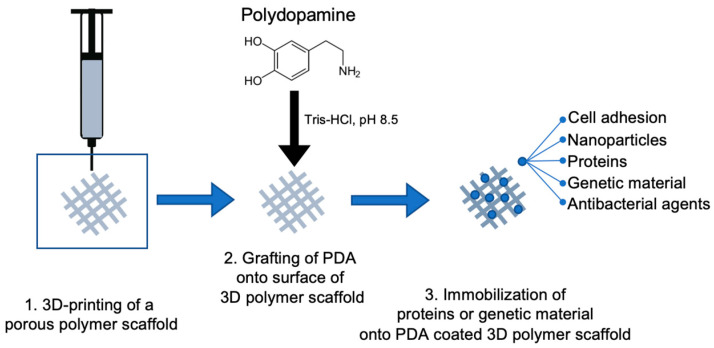
Schematic illustration of a general method in functionalizing a 3D-printed polymer scaffold by grafting polydopamine onto the surface allowing for the immobilization of various key cells, nanoparticles, proteins, genetic materials, or antibacterial agents to adhere firmly prior to and post-implantation to improve the desired therapeutic effect.

**Figure 4 biomimetics-08-00078-f004:**
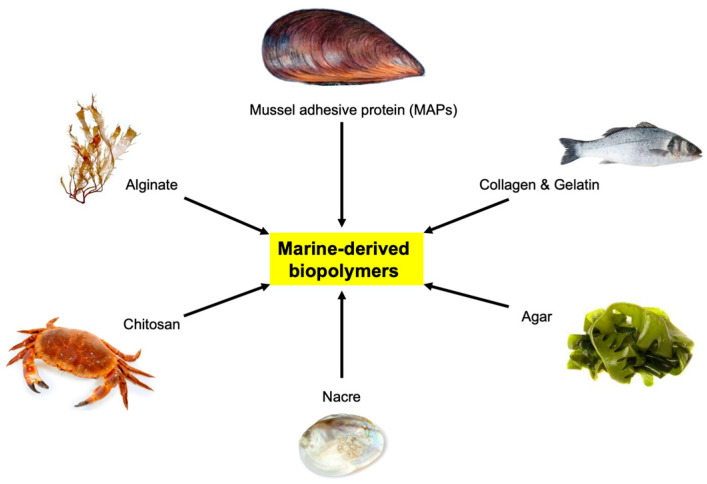
A graphical representation of some of the most commonly researched marine-derived biopolymers that are used within the field of dentistry.

**Table 1 biomimetics-08-00078-t001:** A classification of different biomaterials with examples of their respective applications in the field of dentistry.

Class of Material	Restorative Dentistry	Therapeutic Dentistry	Surgical and Restorative Dentistry
**Metal and alloys**	-Cobalt chromium -Nickel-chromium -Stainless steel -Ti-6Al-4V (Titanium alloy) -Gold -Platinum -Palladium	-Silver nanoparticles -Alloys containing silver -Alloys containing copper	-Stainless steel (316/316L) -Ti-6Al-4V (Titanium alloy) -Cobalt containing alloy (Obsolete)
**Polymers**	-Poly(methyl methylacrylate) (PMMA) -Polyether ether ketone (PEEK) -Urethane Dimethacrylate (UDEM) -Triethylene glycol Dimethacrylate (TEGDMA) -2-hydroxyethyl methacrylate (HEMA)	Polytetrafluoroethylene (PTFE)	-Polyether ether ketone (PEEK) -Polytetrafluoroethylene (PTFE) -Poly(glycolic acid) (PGA) -Polylactic acid (PLA) -Pol(lactic-co-glycolic acid) (PLGA) -Polycaprolactone (PCL) -Polyesters (PE) -Poly(propylene fumarate) (PPE)
**Ceramics**	-Lithium disilicate -Zirconia -Alumina -Leucite -Felspathic porcelain -Micra-based glass ceramics -Spinel-based glass ceramics	-Hydroxyapatite -Beta-tricalcium phosphate	-Hydroxyapatite -Beta-tricalcium phosphate -Alpha-tricalcium phosphate -Zirconia -Alumina -Bioglass -Calcium silicate
**Composites**	-Carbon-fiber reinforced PEEK (CFR-PEEK) -Glass-fibre reinforced PEEK (GFR-PEEK) -Nano/Micro-filled compostite and copomers	Fluoride-releasing copomer	Carbon-fiber reinforced PEEK (CFR-PEEK)

**Table 2 biomimetics-08-00078-t002:** Summary of the advantages and limitations of different 3D printing/additive manufacturing methods and their respective materials that are commonly used in the field of dentistry.

3D-Printing	Advantages	Limitations	Materials	Application in Dentistry
**Stereolithography (SLA)**	-High precision -Cost-effective printing -Smooth post-print finish -Fast printing time	-Weak mechanical properties over long-term -Photosensitivity of materials printed -Requires post-print processing (Wash, cure, drying)	-Ceramic-filled resins -Acrylonitrile butadiene styrene (ABS)—like resins -Polypropylene (PP) like resins	-Temporary prosthetics -Surgical guides -Orthodontic models
**Selective laser melting (SLM)**	-Capable of printing full metal components -Wide range of material choice (metals) -Reduce wastage -High print accuracy	-High temperatures are required -Extensive printing supports -Requires extensive post-printing processing -Expensive and size restriction	-Aluminum alloys -Titanium -Steel -Cobalt chromium -Copper	-Dental crowns and bridges for porcelain fused to metal prostheses -Removable partial dental frameworks
**Digital light processing (DLP)**	-Faster printing time and curing process compared to SLA -High surface quality and accuracy -Cost-effective printing -Wide range of material choice (photopolymers)	-Limited print size -Expensive resin material -Potential warping of larging prints	-Photopolymer resins -Thermoplastic resins -Ultraviolet curing resins -Castable resins	-Patient dental models -Dental implants -Dental bridges -Dental crowns -Bone scaffolds
**Bioprinting**	-Fast printing time -Cost effective printing -High degree in cellular positioning -High print accuracy	-Poor mechanical properties (i.e., scaffolds) -Maintain cell viability during print -Require low viscosity bioinks -Ethical standards	-Chitosan -Hyaluronic acid -Alginate -Collagen -Fibrin/Fibrinogen	-Guided bone/tissue regeneration -Bone and tissue grafts -Cell laden scaffolds for hard and soft dental tissues

## Data Availability

Not applicable.

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
