# Peer review of "Biomimicry and 3D-Printing of Mussel Adhesive Proteins for Regeneration of the Periodontium—A Review"

_biomimetics, 2023, doi:10.3390/biomimetics8010078_

Round 1

Reviewer 1 Report

The part of the paper relating to MAPs (paragraphs 4 and 5) is very interesting and clear.

However, since the authors in the abstract define the paper as a review, which aims to give an overview of biomimetic materials for applications in dentistry, they should significantly revise the first part of the paper (paragraphs 2 and 3)

In paragraph 2 (Biomimetic Biomaterials in Dentistry) the definition of "Biomaterial" is superfluous.

In this paragraphs only biomimetic materials for applications in dentistry should be described (line 120 refers to "biomaterials used in multiple disciplines of medicine"). The classifications of materials based on chemical structure, biocompatibility, origin and hystorically are elementary and unnecessary, however it would be appropriate to describe the specific bimimetic materials applied in dentistry with a greater degree of precision and in-depth analysis, making examples of applications with adequate references and also paying attention to the terms used…..

Some examples

in line 114 "technical, safety and economical" are not properties but specifications

in line 123 the definition of metallic bond should be more precise

in line 130 the information on polymers is not clear….”lower elastic moduli and density “ compared to whom?

In paragraph 3 (Biomimetic Biomaterials in Periodontiun Regeneration) the techniques for the production of scaffolds should be described in a clearer and more precise way by inserting a greater number of references.

Author Response

The part of the paper relating to MAPs (paragraphs 4 and 5) is very interesting and clear.

However, since the authors in the abstract define the paper as a review, which aims to give an overview of biomimetic materials for applications in dentistry, they should significantly revise the first part of the paper (paragraphs 2 and 3)

In paragraph 2 (Biomimetic Biomaterials in Dentistry) the definition of "Biomaterial" is superfluous.

  • We have taken out the definition of “Biomaterial”

In this paragraphs only biomimetic materials for applications in dentistry should be described (line 120 refers to "biomaterials used in multiple disciplines of medicine"). The classifications of materials based on chemical structure, biocompatibility, origin and hystorically are elementary and unnecessary, however it would be appropriate to describe the specific bimimetic materials applied in dentistry with a greater degree of precision and in-depth analysis, making examples of applications with adequate references and also paying attention to the terms used…..

  • We have amended this section and taken out the classification of materials and replaced it with specific examples of biomimetic biomaterials applied in dentistry in more depth explaining their clinical application mainly for the regeneration of the periodontium for example hydroxyapatite, bioactive glass and BMP-2 on lines 114-133.

Some examples

in line 114 "technical, safety and economical" are not properties but specifications

  • “Properties” has been replaced with “specifications” and clarified our explanation on lines 94-95

in line 123 the definition of metallic bond should be more precise

  • We have removed the definition of metallic as it was not relevant to the scope of the review

in line 130 the information on polymers is not clear….”lower elastic moduli and density “ compared to whom?

  • We have removed the information regarding polymers as it was not relevant to the scope of the review

In paragraph 3 (Biomimetic Biomaterials in Periodontiun Regeneration) the techniques for the production of scaffolds should be described in a clearer and more precise way by inserting a greater number of references.

  • More detail on techniques used to produce specific scaffolds for dental applications have been added and we have included more references that refer to those techniques on lines 196-210. We have also changed the order in which the paragraphs are laid out in this section allowing for a clearer understanding of how can scaffolds benefit in periodontium regeneration

Reviewer 2 Report

The subject is timely and of great interest. The authors presented an nice overview of the application of different biomimetic materials in dentistry and others. The analysis and discussion are of in depth. The reviewer believes this is a paper of high quality and it is suggested that the article be accepted after the following minor comments are taken care of.  I will be happy to review the revised manuscript.

1.       Nacre has been used as a model or inspiration material for dental and bone applications. Seashells are made by nature via 3D printing. Many groups have been used nacre-like materials by 3D printing. The authors need to talk about this aspect with reference to the following papers.

Uncovering Aragonite Nanoparticle Self-assembly in Nacre - A Natural Armor, Crystal Growth & Design 12 (2012) 4306-4310

Nanoscale structural and mechanical characterization of a natural nanocomposite material: the shell of red abalone, Nano Letters, 4 (2004) 613-617

Nanoscale structural and mechanical characterization of natural nanocomposites: seashells, JOM, 59 (2007) 71-74

2.       Bioprinting often involves dissimilar materials like biopolymer. Biopolymers have different properties from the hard phase or other materials. The authors need to talk about such aspect with reference to the following paper.

Deformation strengthening of biopolymer in nacre. Advanced Functional Materials, 21 (2011) 3883–3888.

Elastic modulus of biopolymer matrix in nacre measured using coupled atomic force microscopy bending and inverse finite element techniques, Materials Science and Engineering C 31 (2011) 1852-1856

3.       Defects in 3D printed materials are critical to the overall properties and functionalities. It is important to monitor and eliminate defects. The authors may briefly talk about defects in 3D printing with reference to the following paper.

In situ real time defect detection of 3D printed parts, Additive Manufacturing, 17 (2017) 135-142

Author Response

Reviewer 2

The subject is timely and of great interest. The authors presented an nice overview of the application of different biomimetic materials in dentistry and others. The analysis and discussion are of in depth. The reviewer believes this is a paper of high quality and it is suggested that the article be accepted after the following minor comments are taken care of.  I will be happy to review the revised manuscript.

  1. Nacre has been used as a model or inspiration material for dental and bone applications. Seashells are made by nature via 3D printing. Many groups have been used nacre-like materials by 3D printing. The authors need to talk about this aspect with reference to the following papers.

Uncovering Aragonite Nanoparticle Self-assembly in Nacre - A Natural Armor, Crystal Growth & Design 12 (2012) 4306-4310

Nanoscale structural and mechanical characterization of a natural nanocomposite material: the shell of red abalone, Nano Letters, 4 (2004) 613-617

Nanoscale structural and mechanical characterization of natural nanocomposites: seashells, JOM, 59 (2007) 71-74

  • We introduced the biomimetic material nacre on lines 358-364 which specifically relates to a gap in the current literature that we have found on the potential use of nacre-like material within the production of scaffolds vid 3D printing to improve the mechanical properties of scaffolds post-print.
  1. Bioprinting often involves dissimilar materials like biopolymer. Biopolymers have different properties from the hard phase or other materials. The authors need to talk about such aspect with reference to the following paper.

Deformation strengthening of biopolymer in nacre. Advanced Functional Materials, 21 (2011) 3883–3888.

Elastic modulus of biopolymer matrix in nacre measured using coupled atomic force microscopy bending and inverse finite element techniques, Materials Science and Engineering C 31 (2011) 1852-1856

  • The technique of bioprinting using marine-derived biopolymers has been added on lines 348-358, and examples of different marine-derived biopolymers have also been included on lines 354-356.
  • We have also referenced the suggested papers on lines 363-364 that outline the potential of nacre-like materials to be incorporated into the 3D scaffold workflow to overcome some of the limitations (mechanical properties) that current bio-inks and biomaterials in general face when printing a scaffold on lines 365-370.
  1. Defects in 3D printed materials are critical to the overall properties and functionalities. It is important to monitor and eliminate defects. The authors may briefly talk about defects in 3D printing with reference to the following paper.

In situ real time defect detection of 3D printed parts, Additive Manufacturing, 17 (2017) 135-142

  • This comment is very much appreciated, we have gone ahead and added a section within the review paper detailing some of the potential drawbacks of 3D-printed materials. In addition to including the suggested paper, we have also included several other parametric analysis papers that help monitor and eliminate potential defects during the 3D-printing process on lines 299-313.

Round 2

Reviewer 1 Report

Thanks for the changes and corrections that have greatly improved the quality of the paper. The paper in this form may be accepted for publication

Author Response

Your are comments are very much appreciated and we thank you for your time in reviewing our manuscript.